# Trends in HIV incidence between 2013–2019 and association of baseline factors with subsequent incident HIV among gay, bisexual, and other men who have sex with men attending sexual health clinics in England: A prospective cohort study

**Nadia Hanum**[1]*, **Valentina Cambiano**[1], **Janey Sewell**[1], **Alison J. Rodger**[1], **Nneka Nwokolo**[2], **David Asboe**[2], **Richard Gilson**[1,3], **Amanda Clarke**[4], **Ada R. Miltz**[1], **Simon Collins**[5], **Valerie Delpech**[6], **Sara Croxford**[6], **Andrew N. Phillips**[1], **Fiona C. Lampe**[1], **for the AURAH2 Study Group**[¶]

1 UCL Institute for Global Health, London, United Kingdom, 2 Chelsea and Westminster Hospital NHS Foundation Trust, London, United Kingdom, 3 Central and North West London NHS Foundation Trust, London, United Kingdom, 4 Brighton and Sussex University Hospital NHS Trust, Brighton, United Kingdom, 5 HIV i-Base, London, United Kingdom, 6 Public Health England, London, United Kingdom

¶ Membership of the AURAH2 Study Group is provided in the Acknowledgements.
* nadia.hanum.17@ucl.ac.uk

## Abstract

### Background

Prospective cohort studies of incident HIV and associated factors among gay, bisexual, and other men who have sex with men (GBMSM) in the United Kingdom are lacking. We report time trends in and factors associated with HIV incidence between 2013 and 2019 among a cohort of GBMSM: the AURAH2 prospective study.

### Methods and findings

Participants were recruited through 1 of 3 sexual health clinics in London and Brighton (July 2013 to April 2016) and self-completed a baseline paper questionnaire and subsequent 4-monthly and annual online questionnaires (March 2015 to March 2018), including information on sociodemographics, lifestyle, health and well-being, HIV status, sexual/HIV-related behaviours, and preexposure prophylaxis and postexposure prophylaxis (PrEP/PEP). Incident HIV was ascertained by linkage with national HIV surveillance data from Public Health England (PHE). We investigated the associations of HIV incidence with (1) baseline factors using mixed-effects Weibull proportional hazard models, unadjusted and adjusted for age, country of birth and ethnicity, sexuality, and education level; and (2) time-updated factors, using mixed-effects Poisson regression models.

In total, 1,162 men (mean age 34 years, 82% white, 94% gay, 74% university-educated) were enrolled in the study. Thirty-three HIV seroconversions occurred over 4,618.9 person-

**Data Availability Statement:** Any personally identifiable data cannot be made publicly available, because this study was conducted with approval from The National Research Ethics Service (NRES) committee, which requires that to protect participants' privacy data from the studies are released only after they have provided written approval. A de-identified dataset sufficient to reproduce the study findings will be made available upon written request after approval from NRES committee. To request these data, please contact: nres.queries@nhs.net or through www.nres.nhs/contacts/nres-committe-directory/.

**Funding:** The AURAH2 study was funded by the National Institute for Health Research (NIHR) under its Programme Grants for Applied Research Programme (RP-PG-1212-20006). URL funder: https://www.nihr.ac.uk/. AJR holds the grant from the National Institute for Health Research. NH receives funding from the Indonesian Endowment Fund for Education (LPDP Indonesia Scholarship) during the conduct of the study. The funder had no role in study design, data collection and analysis, decision to publish, or preparation of the manuscript.

**Competing interests:** I have read the journal's policy and the authors of this manuscript have the following competing interests: AC reports personal fees from Gilead Sciences for advisory board attendance & conference sponsorship, personal fees from ViiV for advisory board attendance, outside the submitted work. FCL reports grants from National Institute for Health Research (NIHR) during the conduct of the study. NN has been a full-time employee of ViiV Healthcare since March 2019 and reports personal fees, and support for conference attendance from ViiV Healthcare, and Gilead Sciences outside the submitted work. All other authors have declared that no competing interests exist.

**Abbreviations:** ART, antiretroviral therapy; AURAH2, The Attitudes to and Understanding of Risk of Acquisition of HIV 2; CI, confidence interval; CLS, condomless anal sex; GAD-7, Generalised Anxiety Disorder-7; GBMSM, gay, bisexual, and other men who have sex with men; GEE, generalised estimation equation; GP, general practitioner; GUM, genitourinary medicine; HR, hazard ratio; IQR, interquartile range; IR, incidence rate; IRR, incidence rate ratio; LGV, lymphogranuloma venereum; NHS, National Health Service; NRES, National Research Ethics Service; PEP, postexposure prophylaxis; PHE, Public Health England; PHQ-9, Patient Health Questionnaire; PrEP, preexposure prophylaxis; PY, person-years; SD, standard deviation; STI, sexually transmitted

years (PY) of follow-up: an overall HIV incidence rate (IR) of 0.71 (95% confidence interval (CI) 0.51 to 1.00) per 100 PY. Incidence declined from 1.47 (95% CI 0.48 to 4.57) per 100 PY in 2013/2014 to 0.25 (95% CI 0.08 to 0.78) per 100 PY in 2018/2019; average annual decline was 0.85-fold ($p < 0.001$). Baseline factors associated with HIV acquisition included the following: injection drug use (6/38 men who reported injection drug-acquired HIV; unadjusted conditional hazard ratio (HR) 27.96, 95% CI 6.99 to 111.85, $p < 0.001$), noninjection chemsex-related drug use (13/321; HR 6.45, 95% CI 1.84 to 22.64, $p < 0.001$), condomless anal sex (CLS) (26/741; HR 3.75, 95% CI 1.31 to 10·74, $p = 0.014$); higher number of CLS partners (HRs >10 partners [7/57]; 5 to 10 partners [5/60]; and 2 to 4 partners [11/293]: 14.04, 95% CI 4.11 to 47.98; 9.60, 95% CI 2.58 to 35.76; and 4.05, 95% CI 1.29 to 12.72, respectively, $p < 0.001$); CLS with HIV–positive partners (14/147; HR 6.45, 95% CI 3.15 to 13.22, $p < 0.001$), versatile CLS role (21/362; HR 6.35, 95% CI 2.18 to 18.51, $p < 0.001$), group sex (64/500; HR 8.81, 95% CI 3.07 to 25.24, $p < 0.001$), sex for drugs/money (4/55, HR 3.27, 95% CI 1.14 to 9.38, $p = 0.027$) (all in previous 3 months); previous 12-month report of a bacterial sexually transmitted infection (STI) diagnoses (21/440; HR 3.95, 95% CI 1.81 to 8.63, $p < 0.001$), and more than 10 new sexual partners (21/471, HRs 11 to 49, 50 to 99, and >100 new partners: 3.17, 95% CI 1.39 to 7.26; 4.40, 95% CI 1.35 to 14.29; and 4.84, 95% CI 1.05 to 22.4, respectively, $p < 0.001$). Results were broadly consistent for time-updated analysis ($n = 622$ men). The study's main limitation is that men may not be representative of the broader GBMSM population in England.

## Conclusions

We observed a substantial decline in HIV incidence from 2013 to 2019 among GBMSM attending sexual health clinics. Injection drug use, chemsex use, and measures of high-risk sexual behaviour were strongly associated with incident HIV. Progress towards zero new infections could be achieved if combination HIV prevention including Test and Treat strategies and routine commissioning of a PrEP programme continues across the UK and reaches all at-risk populations.

## Author summary

### Why was the study done?

- A decline has been observed in new HIV diagnoses among gay, bisexual, and other men who have sex with men (GBMSM) in the United Kingdom.

- Internationally, an overall decline in HIV diagnoses and incidence among GBMSM has also been reported in several cities in developed countries such as Australia, the United States, the Netherlands, and some other European countries between 2013 and 2019.

- To our knowledge, no prospectively followed cohort studies of GBMSM in England have reported trends in HIV incidence in recent years or on factors associated with incident HIV.

infection; STROBE, Strengthening the Reporting of Observational Studies in Epidemiology.

## What did the researches do and find?

- We estimated trends in HIV incidence between 2013 and 2019 among a cohort of GBMSM attending sexual health clinics, and we found a declining trend.

- We also assessed factors associated with HIV incidence, and our findings emphasise the importance of awareness of high-risk sexual behaviours and recreational drug use (particularly injection drug use and chemsex-associated drug use) as factors associated with HIV acquisition.

- Despite observing significant declines in HIV incidence, incidence rates (IRs) remained high among men who reported injection drugs use, chemsex drug use, condomless sex with multiple partners, and group sex.

## What do these findings mean?

- Growing evidence shows that the HIV transmission declines may potentially be attributed to the comprehensive control and HIV treatment efforts in the UK.

- The continuation of intensification of HIV testing, immediate antiretroviral therapy (ART) initiation, the use of condoms, and routine commissioning of a preexposure prophylaxis (PrEP) programme could potentially ensure that the decline in HIV incidence is felt across all groups impacted by the epidemic.

## Introduction

To bring the HIV epidemic under control, there has been a massive scale-up in the treatment and prevention of HIV over the past decade that has led to a gradual decline in new HIV infections globally [1]. In the United Kingdom (UK), modelling of HIV surveillance data suggests that the underlying incidence of new HIV infections has been falling steadily for more than 5 years (since 2012) [2]. The decline has been particularly marked among gay, bisexual, and other men who have sex with men (GBMSM), among whom 51% of all new HIV diagnoses occurred in the UK in 2018 [3]. In England, the modelled number of incident infections among GBMSM has declined by 65% since 2014, with the most rapid fall after 2016 [3]. The steep declines coincide with a period when increasing numbers of men accessed preexposure prophylaxis (PrEP) [4]. In addition, during this period, there were efforts to increase uptake and frequency of HIV testing, and HIV treatment guidelines changed to recommend prompt initiation of antiretroviral therapy (ART) for people newly diagnosed with HIV. Declines in new HIV diagnoses among GBMSM have also been reported in New South Wales in Australia [5] and San Francisco and New York City in the United States [6,7].

There remains, however, limited data from UK prospective studies assessing HIV acquisition risk, associated factors, and temporal trends for incident HIV [8,9]. Such data could be helpful in providing insight regarding the risk factors driving the HIV epidemic among GBMSM in England. The Attitudes to and Understanding of Risk of Acquisition of HIV 2 (AURAH2) study is among the first prospective observational cohort studies of initially HIV–negative GBMSM in England. We sought to evaluate trends in HIV incidence between 2013 and 2019 and the association of baseline and time-updated demographic, socioeconomic,

health, lifestyle, and behavioural factors with HIV incidence among GBMSM participating in AURAH2.

## Methods

### Study design and participants

Methodological details of the study have been published previously [10]. The AURAH2 study was a prospective cohort study that recruited GBMSM who were HIV negative or of unknown HIV status from 3 large sexual health clinics in London and Brighton (56 Dean Street, London; Mortimer Market Centre, London; and Claude Nicol Clinic, Brighton) from July 2013 to April 2016. Participants were eligible if they were aged 18 years or older and had attended the study clinics for routine testing for sexually transmitted infections (STIs) or HIV. Men were classified as GBMSM for the purposes of the analysis if they met at least one of the following criteria: (i) reported being gay or bisexual; (ii) reported anal sex with a man in the past 3 months; or (iii) reported having disclosed to their family, friends, or workmates as being gay, bisexual, and/or attracted to men. Participants who consented to the study completed a confidential baseline paper questionnaire in the clinic. During the follow-up period, participants self-completed subsequent 4-monthly and annual questionnaires that were available online from March 2015 until March 2018. The baseline questionnaire gathered information on demographic, socioeconomic, lifestyle, health and well-being–related factors, knowledge and understanding of HIV, sexual behaviours, STI diagnoses, and PrEP and postexposure prophylaxis (PEP) use. The 4-monthly questionnaires assessed information on HIV status, HIV testing history, sexual behaviours, and lifestyle factors. Annual questionnaires captured the same information as the 4-monthly questionnaire and additional information on PrEP and PEP use in the past year, relationship status, and health and well-being factors as assessed on the baseline questionnaire. This study is reported as per the Strengthening the Reporting of Observational Studies in Epidemiology (STROBE) guideline (**S1 Checklist**).

### Ethics approval and participant consent

All participants provided written, informed consent before taking part. Consent to participate in the study included consent for linkage to Public Health England (PHE)'s datasets at the end of the study using limited participant identifiers. The AURAH2 study was approved by the designated research ethics committee, The National Research Ethics Service (NRES) committee London-Hampstead, ref: 14/LO/1881 in November 2014 [10]. Based on the research protocol and all versions of study documents, the AURAH2 study subsequently received permission for clinical research at the 3 participating National Health Service (NHS) sites: Chelsea and Westminster NHS Foundation Trust, Central and North West London NHS Foundation Trust, and the Brighton and Sussex University Hospitals NHS Trust. The AURAH2 study was registered on the NIHR clinical research network portfolio.

### Completion of online follow-up questionnaires

Participants who completed a first online follow-up questionnaire in March 2015 had the option to complete up to 9 online questionnaires, as the follow-up finished in March 2018. When participants were due to complete a questionnaire, 2 remainder emails were sent after 2 and 4 weeks followed by a text message. If participants missed a questionnaire at any time during follow-up, they were still invited to complete subsequent questionnaires. At each online follow-up, participants were asked about the most recent date of HIV test and the result.

## Baseline measures

All baseline measures were self-reported in the participant baseline questionnaire. Sociodemographic variables included age group (<25; 25 to 29; 30 to 34; 35 to 39; 40 to 44; ≥45 years), country of birth and ethnicity (white UK born; other ethnicity UK born; white non-UK born; other ethnicity non-UK born), self-reported sexual identity (gay; bisexual/other plurisexual identities), education (university degree; other qualification; no qualification), ongoing relationship (yes, living with partner; yes, not living with partner; no), employment status (employed; not employed), sufficient money for basic needs (yes; mostly; sometimes or no), and housing status (homeowner; renting including private, housing association, and council; unstable or other).

We considered the following measures of sexual/HIV-related behaviour in the preceding 3 months (classified as "yes" or "no" unless otherwise indicated): condomless anal sex (CLS), number of CLS partners (none; 1; 2 to 4; 5 to 10; >10), CLS with partners known to be HIV positive, sexual CLS role (no CLS; always insertive; always receptive; insertive and receptive [versatile]), group sex, sex for drugs or money, fisting, or sex toys use. We also considered HIV test in the previous 6 months, and bacterial STI diagnosis, number of new sexual partners (0 to 10; 11 to 49; 50 to 99; ≥100), and PrEP and PEP use in the previous 12 months. Bacterial STIs included gonorrhoea, syphilis, and chlamydia, including lymphogranuloma venereum (LGV).

Lifestyle factors included recreational drug use (injection drug use; noninjection use of 1 or more of the 3 chemsex-associated drugs [mephedrone, GHB/GBL, crystal methampethamine]; non-injection use of other drugs; no drug use), smoking status (never smoked; ex-smoker; current smoker) and alcohol consumption (higher-risk alcohol consumption: a score of ≥6 on a modified version the AUDIT-C WHO alcohol screening tool questionnaire, first 2 questions only) [11]. A total score of 6 was chosen given that AURAH2 participants were only asked the first 2 questions of the WHO AUDIT-C questionnaire rather than the full AUDIT-C. Mental health included symptoms of depression (defined as a score of ≥10 on the Patient Health Questionnaire [PHQ-9], which is the standard cutoff score used to define clinically significant depressive symptoms) [12], and anxiety symptoms (defined as a score of ≥10 on the Generalised Anxiety Disorder Scale [GAD-7], which represents the standard cutoff to define anxiety disorder) [13].

For sexual/HIV-related behaviour, mental health, and alcohol consumption measures, missing responses were considered to indicate the absence of the event or condition, because our outcome of interests was "past report of behaviours." Where there was no report of these measures, including missing, we classified answers as "no." For all other variables that were not classified as "yes" or "no," missing values were excluded from the analyses.

## Time-updated measures

Age, recent HIV test, CLS, CLS with 2 or more partners, sexual CLS role, group sex, chemsex (a different variable from recreational drug use variable at baseline questionnaire; "have you used drugs before or during sex (chemsex) in the last three months?", classified as "yes" or "no"), and bacterial STI diagnosis were also used as time-varying variables derived from baseline, 4-monthly, and annual questionnaires. Relationship status, PrEP use, PEP use, recreational drug use, injection drug use, depressive symptoms, anxiety symptoms, and alcohol use were time-varying variables derived from baseline and annual questionnaires. All other variables were fixed variables that were only ascertained at baseline.

## Ascertainment of incident HIV

There were 2 methods of ascertainment of incident HIV diagnosis during follow-up. First, records of all GBMSM enrolled in the AURAH2 study were linked to national HIV surveillance data by PHE [14,15]. The databases collect information on new HIV diagnoses from

laboratories, genitourinary medicine (GUM) clinics, general practitioners (GPs), and other services where HIV testing takes place in England. The data linkage process was carried out using a deterministic and hierarchical algorithm, based on gender identity, date of birth, year of birth, country of birth, ethnicity, originating clinic, years in the UK, and first initial and Soundex code (a 4-character coding of an adult surname). All data collected as part of the national HIV surveillance programme in the UK is pseudo-anonymised; no names are collected. The data matching process was completed in November 2019. For each study participant that matched to the HIV surveillance dataset, PHE data were provided on date and region of HIV diagnosis, CD4 and viral load at HIV diagnosis, and if relevant, time from diagnosis to linkage to care, time from diagnosis to treatment initiation, and death.

The second method of ascertainment of new HIV diagnoses was through the online follow-up questionnaires; participants were asked about the date and results of most recent HIV test. All the participants who reported being newly diagnosed with HIV in a follow-up questionnaire were also identified as having a new HIV diagnosis in the PHE surveillance databases. Linkage with PHE databases also identified a small number of participants who were positive at entry to the study ($n = 3$); these men were excluded from analysis.

## Statistical analysis

For the analyses of HIV incidence and baseline-associated factors, all men enrolled in AURAH2 were included. Incident HIV infection was defined as seroconversion from HIV–negative status at baseline to HIV–positive during follow-up, confirmed by PHE. Person-years (PY) of follow-up were calculated from the date of completing the baseline questionnaire until (1) the date of HIV diagnosis from PHE for men who seroconverted or (2) 3 months before the date of data linkage with PHE datasets was completed (June 30, 2019) for men who did not seroconvert. Due to the linkage with PHE data for ascertainment of the endpoint, all men could be considered as remaining under follow-up over the entire period, even if follow-up questionnaires were not completed.

HIV incidence rates (IRs) were calculated as the number of new HIV infections divided by the number of PY of follow-up, reported with 95% confidence intervals (95% CIs). IRs were calculated per 100 PY, overall and according to calendar year from 2013 until 2019. As the study started on July 30, 2013 and ended on June 30, 2019, the first 2 years (2013 to 2014) and the last 2 years (2018 to 2019) were combined. The associations of baseline factors and current calendar year as a continuous variable with HIV incidence were analysed by calculating HIV IRs and using 2-level random-intercept proportional hazard models with sexual health clinic sites defining the second level to estimate conditional hazard ratios (HRs). The conditional distribution of the response given the random effects was assumed to be a Weibull distribution. HRs with 95% CI are presented unadjusted, and adjusted for sociodemographic factors that were less likely to be influenced by HIV and sexual behaviour: age at baseline, country of birth and ethnicity, sexual identity, and education.

Changes in the annual prevalence of sexual/HIV-related risk behaviours over time were also examined. The prevalence of CLS with 2 or more partners, group sex, bacterial STI, any recreational drug use, injection drug use, noninjection chemsex-related drug use (all in the previous 3 months), and PrEP and PEP use in the previous 12 months was calculated for each year from 2013/14 to 2018/19, using all available baseline and follow-up questionnaires from all participants at each time point. Trends over calendar time during the AURAH2 study period were assessed using univariate generalised estimation equation (GEE) models with a logit link and robust standard errors, accounting for multiple questionnaires responses from individual participants.

We also performed an additional longitudinal analysis among men who completed at least 1 online follow-up questionnaire to examine time-updated factors associated with HIV incidence. We used 2-level random-intercept Poisson regression models, unadjusted and adjusted for age (time-updated), country of birth and ethnicity, sexual identity, and education, using all available baseline and follow-up questionnaires. We present these results as incidence rate ratios (IRRs) with their corresponding 95% CI. In the multivariable analyses, the whole statistical unit for a single individual with missing values was excluded from the analyses if a value for one of the covariates was missing (complete case analysis).

All analyses were planned prior to analysing final datasets from PHE in November 2019 (**S1 Analyses Plan**), and no data-driven changes took place to these analyses, except that we used mixed-effects modelling instead of Cox proportional hazard modelling (indicated in the analyses plan), in response to peer review comments. The use of hierarchical models was chosen to take into account of clustering according to clinic. All analyses were conducted using Stata statistical software (version 15.1).

## Results

### Characteristics of the participants

Between July 2013 and April 2016, a total of 1,162 HIV–negative men were enrolled in the study (**Table 1**). At baseline, the mean age of participants was 34 years (standard deviation [SD]: 10.4; interquartile range [IQR]: 26 to 39), 81.9% were of white ethnicity, 93.6% self-reported being gay, 74.4% had a university degree, 82.9% reported being employed, and 77.4% always had money to cover basic needs. In the previous 3 months, 63.9% reported having had CLS, 35.4% reported CLS with 2 or more partners, 12.7% reported CLS with HIV–positive partners, 43.1% reported group sex, 60.0% reported the use of at least 1 recreational drug, and 3.3% reported injection drug use. The type of drug injected was not ascertained in the baseline questionnaire, but all 38 people who injected drugs reported having taken at least 1 chemsex-related drug in the past 3 months. Overall, 38.0% of men reported having been diagnosed with a bacterial STI in the past year, and 5.0% and 20.7% reported ever having taken PrEP and PEP, respectively, in the past year. Three individuals did not complete the baseline questionnaire. The proportion of missing responses was low (<5% for all variables) (see footnotes in **Table 1**).

Of the 1,162 men enrolled, all were included in the PHE linkage for ascertainment of new HIV diagnosis. Of the 1,162 men, 622 completed at least 1 online follow-up questionnaire (54%), of whom 483 (78% of 622) completed at least 1 annual follow-up questionnaire, and 400 men (64% of 622) were followed until the end of the study. Men who were older, had greater financial security, with more stable housing, with university level education, and were employed were more likely to continue on the study (622 men versus 540 men who completed only the baseline questionnaire) (**S1 Table**). The number of follow-up questionnaires (4-monthly and annual) completed by the end of the study period was 3,277. Participants completed a median of 6 (IQR: 3 to 7) online questionnaires.

### Trends in HIV incidence

In total, 33 of 1,162 men (2.8%) were newly diagnosed with HIV during the period from the date of completion of their baseline questionnaire until June 2019. Of all 33 diagnoses identified by the PHE linkage, 15 were self-reported by the participant on one of the AURAH2 online follow-up questionnaires. There were no additional unconfirmed self-reported HIV diagnoses. The 3 men who did not complete a baseline questionnaire were included in the incidence analysis as data on their age, HIV status, and PY of follow-up time were available from PHE. There were no deaths recorded among the 33 men diagnosed with HIV.

**Table 1. Baseline characteristics and association with incident HIV among 1,162 GBMSM participating in the AURAH2 prospective study, 2013–2019\*.**

| Baseline characteristics | Participants N (%) | HIV infections from baseline– 2019 n (%) | PY at risk | HIV IR per 100 PY (95% CI) | Unadjusted conditional HR (95% CI) | p-value |
|---|---|---|---|---|---|---|
| **Demographic characteristics** | | | | | | |
| **Age at baseline category, years** | | | | | | 0.421 0.417[t] |
| <25 | 275 (23.9) | 8 (2.9) | 1087.61 | 0.74 (0.37–1.47) | 1 (Ref) | |
| 25–29 | 207 (17.9) | 3 (1.5) | 839.50 | 0.36 (0.11–1.10) | 0.49 (0.13–1.85) | |
| 30–34 | 227 (19.7) | 5 (2.2) | 896.65 | 0.56 (0.23–1.34) | 0.76 (0.25–2.32) | |
| 35–39 | 156 (13.5) | 8 (5.1) | 605.63 | 1.32 (0.66–2.64) | 1.79 (0.67–4.76) | |
| 40–44 | 121 (10.5) | 4 (3.3) | 480.92 | 0.83 (0.31–2.22) | 1.13 (0.34–3.77) | |
| ≥45 | 167 (14.5) | 5 (2.9) | 674.09 | 0.74 (0.31–1.78) | 1.02 (0.33–3.12) | |
| **Mean age (SD)** | **34 (10.4)** | | | | | |
| **Median age (IQR)** | **31 (26–39)** | | | | | |
| **Country of birth and ethnicity§** | | | | | | 0.176 |
| Born in the UK, white | 568 (49.4) | 10 (1.8) | 2,296.31 | 0.44 (0.23–0.81) | 1 (Ref) | |
| Born in the UK, other ethnicity | 60 (5.2) | 1 (1.7) | 242.16 | 0.41 (0.06–2.93) | 0.94 (0.12–7.38) | |
| Non-UK born, white | 374 (32.5) | 17 (4.5) | 1,463.62 | 1.16 (0.23–1.34) | 2.63 (1.21–5.76) | |
| Non-UK born, other ethnicity | 148 (12.9) | 2 (1.4) | 581.72 | 0.34 (0.66–2.64) | 0.78 (0.17–3.54) | |
| **Sexual identity** | | | | | | 0.128 |
| Gay | 1,076 (93.6) | 26 (2.4) | 4,291.83 | 0.61 (0.41–0.89) | 1 (Ref) | |
| Bisexual/other | 74 (6.4) | 4 (5.4) | 291.26 | 1.37 (0.52–3.66) | 2.26 (0.79–6.49) | |
| **Socioeconomic characteristics and partnership status** | | | | | | |
| **Education** | | | | | | **0.014 0.013[t]** |
| University degree | 853 (74.4) | 17 (1.9) | 3,413.05 | 0.49 (0.31–0.80) | 1 (Ref) | |
| Other qualification | 272 (23.8) | 11 (4.4) | 1,108.62 | 1.02 (0.56–1.84) | 2.01 (0.94–4.28) | |
| No qualification | 21 (1.8) | 2 (9.5) | 75.83 | 2.64 (0.66–10.55) | **4.65 (1.07–20.14)** | |
| **Employed†** | | | | | | 0.074 |
| Yes | 952 (82.9) | 29 (3.1) | 3,767.14 | 0.77 (0.53–1.10) | 1 (Ref) | |
| No | 197 (17.1) | 1 (0.5) | 812.17 | 0.12 (0.01–0.87) | 0.16 (0.02–1.19) | |
| **Money to cover basic needs** | | | | | | 0.627 0.613[t] |
| All of the time | 896 (77.4) | 24 (2.7) | 3,581.41 | 0.67 (0.45–1.00) | 1 (Ref) | |
| Most of the time | 194 (16.8) | 5 (2.6) | 768.90 | 0.65 (0.27–1.56) | 0.97 (0.37–2.54) | |
| Sometimes/No | 68 (5.9) | 1 (1.5) | 264.84 | 0.38 (0.05–2.68) | 0.55 (0.07–4.09) | |
| **Housing status ᵃ** | | | | | | 0.342 0.330[t] |
| Renting | 680 (59.3) | 13 (1.9) | 2,707.24 | 0.48 (0.28–8.27) | 1 (Ref) | |
| Home owner | 314 (27.4) | 14 (4.5) | 1,252.75 | 1.11 (0.66–1.89) | 2.34 (1.10–4.97) | |
| Unstable or other | 153 (13.3) | 3 (1.9) | 611.33 | 0.49 (0.16–1.52) | 1.02 (0.29–3.58) | |
| **Ongoing relationship** | | | | | | 0.200 0.191[t] |
| Yes, living with partner | 272 (23.5) | 11 (4.0) | 1,080.22 | 1.01 (0.56–1.84) | 1 (Ref) | |
| Yes, not living with partner | 193 (16.7) | 3 (1.6) | 783.15 | 0.38 (0.12–1.19) | 0.38 (0.10–1.35) | |
| No | 693 (59.8) | 16 (2.3) | 2,755.05 | 0.58 (0.36–0.95) | 0.57 (0.26–1.22) | |
| **Sexual/HIV-related behaviour characteristics** | | | | | | |
| **HIV test in the past 6 months** | | | | | | 0.325 |
| No | 322 (27.8) | 6 (1.9) | 1,324.99 | 0.45 (0.20–1.01) | 1 (Ref) | |

(*Continued*)

**Table 1.** (*Continued*)

| Baseline characteristics | Participants *N* (%) | HIV infections from baseline– 2019 *n* (%) | PY at risk | HIV IR per 100 PY (95% CI) | Unadjusted conditional HR (95% CI) | *p*-value |
|---|---|---|---|---|---|---|
| Yes | 837 (72.2) | 24 (2.9) | 3,293.42 | 0.73 (0.49–1.09) | 1.57 (0.64–3.85) | |
| **CLS in the past 3 months**[‡] | | | | | | **0.014** |
| No | 418 (36.1) | 4 (0.9) | 1,704.76 | 0.23 (0.09–0.63) | **1 (Ref)** | |
| Yes | 741 (63.9) | 26 (3.5) | 2,913.66 | 0.89 (0.61–1.31) | **3.75 (1.31–10.74)** | |
| **Number of CLS partners in the past 3 months**[‡] | | | | | | **<0.001** **<0.001** **[t]** |
| No CLS partners | 424 (36.6) | 4 (0.9) | 1,727.09 | 0.23 (0.09–0.62) | 1 (Ref) | |
| One CLS partner | 325 (28.0) | 3 (0.9) | 1,306.35 | 0.23 (0.07–0.71) | 0.98 (0.22–4.40) | |
| 2–4 CLS partners | 293 (25.3) | 11 (3.8) | 1,163.99 | 0.95 (0.52–1.71) | **4.05 (1.29–12.72)** | |
| 5–10 CLS partners | 60 (5.2) | 5 (8.3) | 212.67 | 2.36 (0.98–5.64) | **9.60 (2.58–35.76)** | |
| More than 10 CLS partners | 57 (4.9) | 7 (12.3) | 208.30 | 3.36 (1.60–7.05) | **14.04 (4.11–47.98)** | |
| **CLS with partners known to be HIV positive in the past 3 months**[‡] | | | | | | **<0.001** |
| No | 1,012 (87.3) | 16 (1.6) | 4,086.01 | 0.39 (0.24–0.64) | **1 (Ref)** | |
| Yes | 147 (12.7) | 14 (9.5) | 532.41 | 2.63 (1.56–4.44) | **6.45 (3.15–13.22)** | |
| **Sexual role CLS in the past 3 months** | | | | | | **<0.001** |
| No CLS/did not state which partner | 423 (36.5) | 4 (0.9) | 1,724.83 | 0.23 (0.08–0.62) | 1 (Ref) | |
| Always insertive | 217 (18.7) | 2 (0.9) | 877.05 | 0.22 (0.05–0.91) | 0.98 (0.18–5.35) | |
| Always receptive | 157 (13.6) | 3 (1.9) | 623.35 | 0.48 (0.15–1.49) | 2.06 (0.46–9.19) | |
| Versatile (sometimes insertive, sometimes receptive) | 362 (31.2) | 21 (5.8) | 1,393.19 | 1.51 (0.9–2.31) | **6.35 (2.18–18.51)** | |
| **Number of new sexual partners in the past 12 months~** | | | | | | **0.001** **0.001[t]** |
| 0–10 new partners | 688 (59.4) | 9 (1.3) | 2,772.34 | 0.32 (0.17–0.62) | 1 (Ref) | |
| 11–49 new partners | 367 (31.6) | 15 (4.1) | 1,446.30 | 1.04 (0.63–1.72) | **3.17 (1.39–7.26)** | |
| 50–99 new partners | 72 (6.2) | 4 (5.6) | 272.66 | 1.47 (0.55–3.91) | **4.40 (1.35–14.29)** | |
| 100 or more new partners | 32 (2.8) | 2 (6.3) | 127.12 | 1.57 (0.39–6.29) | **4.84 (1.05–22.41)** | |
| **Group sex in the past 3 months** | | | | | | **<0.001** |
| No | 659 (56.9) | 4 (0.6) 64 | 2,670.75 | 0.15 (0.06–0.39) | 1 (Ref) | |
| Yes | 500 (43.1) | 64 (12.8) | 1,947.67 | 1.33 (0.91–1.96) | **8.81 (3.07–25.24)** | |
| **Fisting or sex toys use in the past 3 months** | | | | | | 0.202 |
| No | 745 (64.3) | 16 (2.2) | 2,982.81 | 0.54 (0.33–0.88) | 1 (Ref) | |
| Yes | 414 (35.7) | 14 (3.4) | 1,635.61 | 0.86 (0.51–1.45) | 1.59 (0.77–3.25) | |
| **Sex for drugs or money in the past 3 months** | | | | | | **0.027** |
| No | 1,104 (95.2) | 26 (2.4) | 4,418.16 | 0.59 (0.40–0.86) | 1 (Ref) | |
| Yes | 55 (4.8) | 4 (7.3) | 200.26 | 1.99 (0.75–5.32) | **3.27 (1.14–9.38)** | |
| **PEP use in the past 12 months** | | | | | | **0.029** |
| No | 919 (79.3) | 19 (2.1) | 3,709.52 | 0.51 (0.33–0.80) | 1 (Ref) | |
| Yes | 240 (20.7) | 11 (4.6) | 908.89 | 1.21 (0.67–2.18) | **2.29 (1.09–4.81)** | |
| **PrEP use in the past 12 months** | | | | | | 0.190 |
| No | 1,101 (95) | 27 (2.7) | 4,408.52 | 0.61 (0.42–0.89) | 1 (Ref) | |
| Yes | 58 (5.0) | 3 (5.2) | 209.49 | 1.43 (0.46–4.44) | 2.21 (0.67–7.30) | |
| **Bacterial STI diagnoses in the past 12 months** | | | | | | **0.001** |
| No | 719 (62.0) | 9 (1.3) | 2,936.07 | 0.31 (0.16–0.59) | 1 (Ref) | |

(*Continued*)

**Table 1.** (Continued)

| Baseline characteristics | Participants N (%) | HIV infections from baseline– 2019 n (%) | PY at risk | HIV IR per 100 PY (95% CI) | Unadjusted conditional HR (95% CI) | p-value |
|---|---|---|---|---|---|---|
| Yes | 440 (38.0) | 21 (4.8) | 1,682.35 | 1.25 (0.81–1.91) | **3.95 (1.81–8.63)** | |
| **Health and lifestyle characteristics** | | | | | | |
| **Smoking status** | | | | | | 0.735 |
| Never smoked | 612 (53.1) | 14 (2.3) | 2,452.23 | 0.57 (0.34–0.96) | 1 (Ref) | |
| Ex-smoker | 290 (25.2) | 8 (2.8) | 1,163.82 | 0.69 (0.34–1.37) | 1.20 (0.50–2.87) | |
| Regular smoker | 250 (21.7) | 8 (3.2) | 977.79 | 0.82 (0.41–1.64) | 1.41 (0.59–3.37) | |
| **Recreational drug use in the past 3 months** | | | | | | **<0.001** **<0.001** [t] |
| No | 464 (40.0) | 3 (0.7) | 1,895.32 | 0.16 (0.05–0.49) | 1 (Ref) | |
| Noninjection drug and non-chemsex use | 336 (29.0) | 8 (2.4) | 1,350.96 | 0.59 (0.29–1.18) | 3.73 (0.99–14.05) | |
| Chemsex-related drug use (no injection) | 321 (27.7) | 13 (4.1) | 1,254.47 | 0.97 (0.61–1.79) | **6.45 (1.84–22.64)** | |
| Injection drug use | 38 (3.3) | 6 (15.8) | 126.67 | 4.74 (2.13–10.54) | **27.96 (6.99–111.85)** | |
| **Higher-risk alcohol consumption (modified WHO AUDIT-C score of ≥6)** | | | | | | 0.714 |
| No | 935 (80.1) | 25 (2.7) | 3,721.68 | 0.67 (0.45–0.99) | 1 (Ref) | |
| Yes | 224 (19.3) | 5 (2.2) | 896.74 | 0.56 (0.23–1.34) | 0.83 (0.32–2.17) | |
| **Depressive symptoms (PHQ-9 score ≥10)** | | | | | | 0.844 |
| No | 1,018 (87.8) | 26 (2.6) | 4,064.75 | 0.64 (0.43–0.93) | 1 (Ref) | |
| Yes | 141 (12.2) | 4 (2.8) | 553.67 | 0.72 (0.27–1.92) | 1.12 (0.39–3.20) | |
| **Anxiety symptoms (GAD-7 score ≥10)** | | | | | | 0.462 |
| No | 1,033 (89.1) | 28 (2.7) | 4,118.39 | 0.68 (0.47–0.98) | 1 (Ref) | |
| Yes | 126 (10.9) | 2 (1.6) | 500.03 | 0.39 (0.10–1.59) | 0.58 (0.14–2.45) | |
| **Year of enrolment** | | | | | | 0.430 |
| 2013 | 28 (2.4) | 2 (7.1) | 149.98 | 1.33 (0.33–5.33) | 1 (Ref) | |
| 2014 | 152 (13.1) | 3 (1.9) | 735.62 | 0.4 (0.13–1.26) | 0.29 (0.05–1.94) | |
| 2015 | 788 (67.8) | 21 (2.7) | 3,115.90 | 0.67 (0.44–1.03) | 0.43 (0.09–1.93) | |
| 2016 | 194 (16.7) | 7 (3.6) | 617.36 | 1.13 (0.54–2.38) | 0.67 (0.11–4.09) | |

*All measures were self-reported, missing data, or missing questionnaire for:

Age: 9 (all HIV negative); Country of birth and ethnicity, Sexuality: 12 (9 HIV negative, 3 HIV positive); University education: 16 (13 HIV negative, 3 HIV positive); Relationship status, Money status: 4 (1 HIV negative, 3 HIV positive); Employment: 13 (10 HIV negative, 3 HIV positive); Housing status: 15 (12 HIV negative, 3 HIV positive); Smoking status: 10 (7 HIV negative, 3 HIV positive); HIV test, CLS, Number of CLS partners, New sexual partners, Sexual CLS role, Group sex, Fisting or sex toys use, PEP use, PrEP use, Recreational drug use, STI diagnoses, Alcohol consumption, Depressive symptoms, and Anxiety symptoms: 3 (all HIV positive).

[t] p-value for trend.

§Other ethnicity includes black, Asian, mixed, and other ethnic group.

†Employed group includes full-time (n = 845) and part-time (n = 107) employment/self-employment; No employment group includes unemployed registered or not registered for benefits (n = 60), sick or disabled (n = 6), retired (n = 24), and other (student or training or looking after home or dependents or other) (n = 107).

ᵉRenting housing includes private renting and renting from council or housing association; unstable or other housing includes temporary accommodation, staying with friends or family, other accommodation, and homeless.

‡CLS with men only.

˜New partners include men and women.

AURAH2, The Attitudes to and Understanding of Risk of Acquisition of HIV 2; CI, confidence interval; CLS, condomless anal sex; GAD-7, generalised anxiety disorder-7; GBMSM, gay, bisexual, and other men who have sex with men; HR, hazard ratio; IQR, interquartile range; IR, incidence rate; PEP, postexposure prophylaxis; PrEP, preexposure prophylaxis; PY, person-years; PHQ-9, patient health questionnaire-9; SD, standard deviation; STI, sexually transmitted infection; WHO-AUDIT, World Health Organization–Alcohol Use Disorders Identification Test.

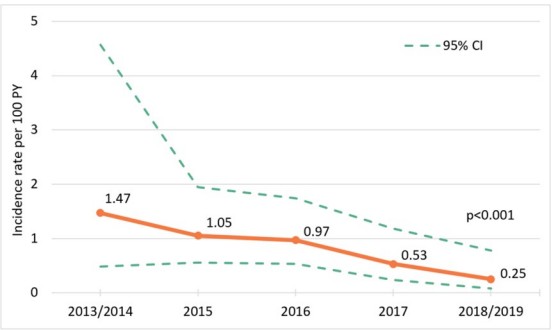

**Fig 1. HIV incidence among GBMSM in the AURAH2 study, 2013–2019.** AURAH2, The Attitudes to and Understanding of Risk of Acquisition of HIV 2; CI, confidence interval; GBMSM, gay, bisexual, and other men who have sex with men; PY, person-years.

The overall HIV IR in this cohort with 4,618.9 PY of follow-up time was 0.71 (95% CI 0.51 to 1.00) per 100 PY (**Fig 1** and **Table 2**). HIV incidence fell progressively from 2013 until 2019; from 1.47 (95% CI 0.48 to 4.57) per 100 PY in 2013/2014 to 0.25 (95% CI 0.08 to 0.78) per 100 PY in 2018/2019. The incidence declined on average by 0.85-fold per year from 2013 to 2019 ($p < 0.001$, modelled using mixed-effects Weibull proportional hazard).

The most common age category at the time of new HIV diagnoses was between 35 and 44 years, with a total of 13 men (39.4%) in this age range being diagnosed with HIV, followed by men in the age category of 25 to 34 years (8 men, 24.2%), <25 years (7 men, 21.2%), and ≥45 years (5 men, 15.2%). The median (IQR) age at time of new HIV diagnosis was 35 years (26 to 40).

## Association of baseline factors with incident HIV

Table 1 presents the association of baseline factors with incident HIV diagnosis. In univariable mixed-effects Weibull proportional hazard models, the factor most strongly associated with HIV acquisition was reporting injection drug use in the past 3 months, with an almost 28-fold higher rate compared to men who did not report recreational drug use (HR 27.96, 95% CI 6.99 to 111.85, global $p < 0.001$). The HIV IR among people who injected drugs was 4.74 (95% CI 2.13 to 10.54) per 100 PY. Having used at least 1 noninjection chemsex-related drug was also strongly associated with HIV acquisition (HR 6.45, 95% CI 1.84 to 22.64, compared to no drug use); the association with non-chemsex–related drugs was weaker (HR 3.73, 95% CI 0.99 to 14.05).

Other sexual/HIV-related behaviour risk factors were strongly associated with increased risk of HIV infection: CLS (HR 3.75, 95% CI 1.31 to 10·74, $p = 0.014$), greater number of CLS

**Table 2. HIV incidence among GBMSM participating in the AURAH2 prospective study, 2013–2019.**

| Calendar year | PY | No. of HIV infections | IR (per 100 PY) | 95% CI |
|---|---|---|---|---|
| 2013/2014 | 203.55 | 3 | 1.47 | 0.48–4.57 |
| 2015 | 953.53 | 10 | 1.05 | 0.56–1.95 |
| 2016 | 1,139.29 | 11 | 0.97 | 0.53–1.74 |
| 2017 | 1,134.80 | 6 | 0.53 | 0.24–1.18 |
| 2018/2019 | 1,187.69 | 3 | 0.25 | 0.08–0.78 |
| Overall | **4,618.86** | **33** | **0.71** | **0.51–1.00** |

AURAH2, The Attitudes to and Understanding of Risk of Acquisition of HIV 2; CI, confidence interval; GBMSM, gay, bisexual, and other men who have sex with men; IR, incidence rate; PY, person-years.

partners, with increased risk for those having at least 2 partners (HR for 2 to 4 partners 4.05, 95% CI 1.29 to 12.72; HR for 5 to 10 partners 9.60, 95% CI 2.58 to 35.76, HR for more than 10 partners 14.05, 95% CI 4.11 to 47.98, compared with no CLS, global $p < 0.001$), CLS with HIV–positive partners (HR 6.45, 95% CI 3.15 to 13.22, $p < 0.001$), versatile CLS role (HR 6.35, 95% CI 2.18 to 18.51, $p < 0.001$), group sex (HR 8.81, 95% CI 3.07 to 25.24, $p < 0.001$), and sex for drugs or money (HR 3.27, 95% CI 1.14 to 9.38, $p = 0.027$) in the past 3 months; reporting a bacterial STI diagnosis in the past 12 months (HR 3.95, 95% CI 1.81 to 8.63, $p = 0.001$), reporting more than 10 new sexual partners in the past 12 months (HR for 11 to 49 new partners 3.17, 95% CI 1.39 to 7.26, HR for 50 to 99 new partners 4.40, 95% CI 1.35 to 14.29, HR for 100 or more new partners 4.84, 95% CI 1.05 to 22.41, compared to 0 to 10 new partners, global $p = 0.001$) and having used PEP in the past 12 months (HR 2.29, 95% CI 1.09 to 4.81, $p = 0.029$).

For socioeconomic and demographic characteristics, lower level of education was associated with increased risk of HIV infection (HR for no qualification 4.65, 95% CI 1.07 to 20.14 compared to university degree, global $p = 0.014$). There was some evidence that nonemployed men were at lower risk of infection than employed men (HR 0.16, 95% CI 0.02 to 1.19, $p = 0.074$).

Adjustment for age at baseline, country of birth and ethnicity, sexual identity, and education did not materially change the associations between incident HIV and baseline factors (S2 Table). There were no significant associations of age group, housing status, financial status, relationship status, HIV test in the past 6 months, fisting or sex toys use in the past 3 months, PrEP use in the past 12 months, smoking status, alcohol consumption, country of birth and ethnicity, sexual identity, year of enrolment, depressive symptoms, and anxiety symptoms at baseline with risk of HIV infection (Table 1).

## Prevalence of sexual risk behaviours over time

Fig 2 shows the trends in reported sexual risk behaviours, drug use, and the use of PrEP and PEP by calendar year, based on all available baseline and follow-up questionnaires from all 1,162 participants enrolled (total 4,439 questionnaires). Fig 2A shows that the annual prevalence of CLS with 2 or more partners in the past 3 months increased somewhat from 38.3% to 41.0% (p-value for linear trend from GEE logistic model = 0.006) between 2013/2014 and 2018, while group sex declined substantially from 46.7% to 24.2% ($p < 0.001$), as did bacterial

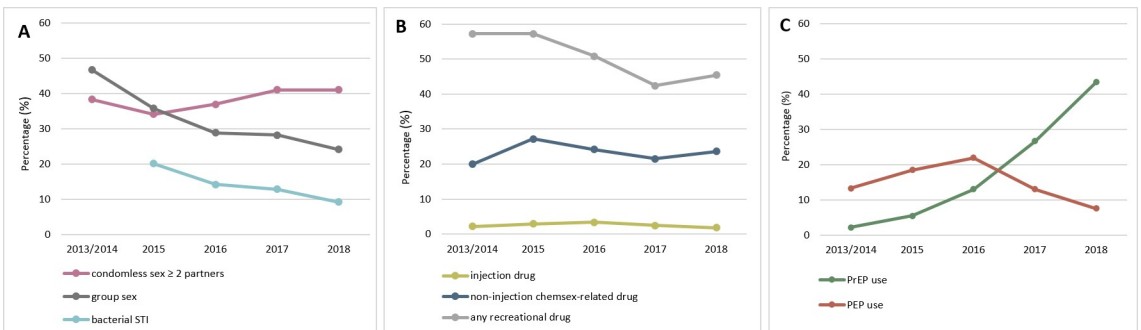

**Fig 2. Annual prevalence of sexual/HIV-related behaviours among GBMSM in the AURAH2 study, 2013–2018\*.** \*Annual reports of (A) sexual risk behaviours in the previous 3 months, data from all available baseline, 4-monthly, and annual questionnaires ($N$ = 4,439 questionnaires), bacterial STI in the previous 3 months, data from 4-monthly and annual questionnaires ($N$ = 3,277 questionnaires); (B) recreational drug use in the past 3 months, data from baseline and annual questionnaires ($N$ = 2,104 questionnaires); (C) PrEP and PEP use in the last 12 months, data from baseline and annual questionnaires ($N$ = 2,085 questionnaires). AURAH2, The Attitudes to and Understanding of Risk of Acquisition of HIV 2; GBMSM, gay, bisexual, and other men who have sex with men; PEP, postexposure prophylaxis; PrEP, preexposure prophylaxis; STI, sexually transmitted infection.

STI diagnoses in the last 3 months from 20.1% to 9.2% ($p < 0.001$) between 2015 and 2018. For bacterial STIs, data were not included from the baseline questionnaire (since 2013) as it asked about diagnoses in the previous 12 months rather than in the last 3 months. Any recreational drug use in the past 3 months decreased from 57.2% to 45.5% ($p < 0.001$), while injection drug use (prevalence around 2%, $p = 0.903$) and the use of at least 1 chemsex-related drug (prevalence between 20% and 30%, $p = 0.232$) were stable (**Fig 2B**). Past 12-month PrEP use increased significantly from 2.22% to 43.4% ($p < 0.001$); on the other hand, PEP use peaked at 21.9% in 2016, then declined to 7.6% in 2018 ($p = 0.07$) (**Fig 2C**).

## Association of time-updated factors with incidence HIV among men who completed at least one online follow-up questionnaire

Among the 622 men who completed an online follow-up questionnaire, 19 were diagnosed with HIV during the period from the date of completion of their baseline questionnaire until June 30, 2019. With a total of 2,495 PY of follow-up time, the overall HIV IR in this subgroup of men was 0.76 (95% CI 0.49 to 1.19) per 100 PY, similar to the overall IR among all men enrolled in the AURAH2 cohort (0.71, 95% CI 0.51 to 1.00 per 100 PY). **Table 3** shows unadjusted and adjusted IRRs from mixed-effects Poisson models for factors associated with HIV incidence among these men (total complete observations 3,821 questionnaires). In this analysis, age, partnership status, sexual/HIV-related behaviours, PrEP and PEP use, and health and lifestyle variables were time updated, whereas ethnicity and country of birth, education, employment, sexual identity, financial status, and housing status were fixed variables that were only asked at baseline questionnaires. Longitudinal factors associated with HIV incidence among these men were quite similar to those among the 1,162 men, in particular, injection drug use (unadjusted IRR 21.67, 95% CI 3.96 to 118.30, $p < 0.001$), chemsex (3.89, 95% CI 1.35 to 11.22, $p = 0.012$), CLS with 2 or more partners, versatile CLS role, group sex (all in the previous 3 months), bacterial STI diagnosis (in the previous 12 months at the baseline questionnaire and in the past 3 months at the 4-monthly and annual questionnaires), and calendar year.

## Discussion

Using a prospectively followed cohort of initially HIV–negative GBMSM in London and Brighton, we demonstrate a substantial decline in HIV incidence, from 1.47 per 100 PY to 0.25 per 100 PY between 2013 and 2019. The results of an earlier report from England's national STI surveillance system also estimated that the annual HIV incidence among men who have sex with men attending English sexual health clinics decreased from 1.90 per 100 PY in 2012/2013 to 0.79 per 100 PY in 2016/2017 [4]. Based on the CD4 back-calculation model that is used to estimate HIV incidence among GBMSM living in England based on data on new HIV diagnoses, incidence begun to fall in 2012 [2,16].

The substantial decline in HIV incidence in our cohort was also described in some other countries [5–7,17,18]. It may be attributed to important behavioural changes within GBMSM populations. The dramatic decline in HIV infection rates in AURAH2 coincides with declines in the proportion of individuals reporting group sex and any recreational drug use since 2013, and diagnosis of bacterial STIs since 2015. The declining trends in group sex and diagnosis of bacterial STIs have been reported previously among men in AURAH2 who completed at least an online follow-up ($n = 622$), during the online follow-up period (2015 to 2018) [19]. This decline could be a feature of the fact that the study recruited GBMSM attending sexual health clinics for STI testing. Engagement in care for STI monitoring may have had a preventive impact on subsequent STI occurrence, or a

**Table 3. Association of time-updated factors with incident HIV among 622 GBMSM who completed at least 1 online follow-up questionnaire, 2013–2018[*].**

| | Unadjusted IRR (95% CI) | *p*-value | Adjusted[€] IRR (95% CI) | *p*-value |
|---|---|---|---|---|
| **Demographic characteristics** | | | | |
| **Age (time-updated) per year** | 3,785 obs | 0.843 | 3,770 obs | 0.4 |
| | 1.00 (0.96–1.05) | | 1.02 (0.97–1.06) | |
| **Age (time-updated) category** | 3,785 obs | 0.676 | 3,770 obs | 0.325 |
| <25 | Ref | | Ref | |
| 25–29 | 0.22 (0.02–1.93) | | 0.65 (0.21–1.92) | |
| 30–34 | 0.41 (0.07–2.23) | | 0.61 (0.22–1.97) | |
| 35–39 | 0.72 (0.16–3.23) | | 0.99 (0.19–5.17) | |
| 40–44 | 0.35 (0.04–3.14) | | 0.49 (0.05–4.92) | |
| ≥45 | 0.86 (0.23–3.21) | | 1.21 (0.28–5.21) | |
| **Country of birth and ethnicity** | 3,782 obs | 0.953 | 3,770 obs | 0.905 |
| Born in the UK, white | Ref | | Ref | |
| Born in the UK, other ethnicity | 1.63 (0.20–13.29) | | 1.77 (0.21–15.04) | |
| Non-UK born, white | 0.97 (0.28–3.31) | | 1.00 (0.29–3.46) | |
| Non-UK born, other ethnicity | 0.74 (0.09–6.03) | | 0.79 (0.09–6.54) | |
| **Sexual identity** | 3,793 obs | 0.639 | 3,770 obs | 0.569 |
| Gay | Ref | | Ref | |
| Bisexual/other | 1.63 (0.21–12.54) | | 1.82 (0.23–14.34) | |
| **Socioeconomic characteristics and partnership status** | | | | |
| **University education** | 3,805 obs | 0.923 | 3,770 obs | 0.932 |
| Yes | Ref | | Ref | |
| No | 0.94 (0.29–3.06) | | 0.95 (0.29–3.09) | |
| **Employed** | 3,772 obs | 0.305 | 3,760 obs | 0.243 |
| Yes | Ref | | Ref | |
| No | 0.34 (0.04–2.64) | | 0.29 (0.03–2.32) | |
| **Money to cover basic needs** | 3,805 obs | 0.744 | 3,770 obs | 0.829 |
| All of the time | Ref | | Ref | |
| Most of the time | 1.29 (0.29–5.83) | | 1.43 (0.30–7.06) | |
| Sometimes/No | 0.10 (0.05–0.18) | | 0.12 (0.05–0.25) | |
| **Housing status** | 3,750 obs | 0.611 | 3,738 obs | 0.701 |
| Home owner | Ref | | Ref | |
| Renting | 0.56 (0.17–1.85) | | 0.66 (0.16–2.71) | |
| Unstable/other | 0.97 (0.19–4.82) | | 1.12 (0.18–6.81) | |
| **Ongoing relationship[**]** | 1,536 obs | 0.094 | 1,522 obs | 0.148 |
| Yes | Ref | | Ref | |
| No | 0.63 (0.36–1.08) | | 0.65 (0.36–1.17) | |
| **Sexual/HIV-related behaviour characteristics** | | | | |
| **Recent HIV test[†]** | 3,699 obs | 0.34 | 3,651 obs | 0.329 |
| No | Ref | | Ref | |
| Yes | 1.87 (0.51–6.80) | | 1.90 (0.52–6.92) | |
| **CLS in the past 3 months** | 3,821 obs | 0.871 | 3,770 obs | 0.196 |
| No | Ref | | Ref | |
| Yes | 1.09 (0.38–3.14) | | 2.71 (0.60–12.23) | |
| **CLS with 2 or more partners** | 3,819 obs | **0.005** | 3,770 obs | **0.004** |
| One/none | Ref | | Ref | |

(*Continued*)

**Table 3.** (Continued)

| | Unadjusted IRR (95% CI) | *p*-value | Adjusted$^{\in}$ IRR (95% CI) | *p*-value |
|---|---|---|---|---|
| 2 or more | **6.19 (1.72–22.17)** | | **9.39 (2.07–42.66)** | |
| **Sexual role CLS in the past 3 months** | 3,803 obs | 0.705 | 3,752 obs | **0.016** |
| No CLS/did not state which partner | Ref | | Ref | |
| Always insertive | - | | - | |
| Always receptive | 0.95 (0.18–4.88) | | 2.47 (0.35–17.67) | |
| Versatile (sometimes insertive, sometimes receptive) | 1.79 (0.60–5.32) | | **4.55 (1.01–21.11)** | |
| **Group sex in the past 3 months** | 3,819 obs | | 3,770 obs | |
| No | Ref | **0.043** | Ref | **0.029** |
| Yes | **2.98 (1.03–8.61)** | | **3.51 (1.14–10.77)** | |
| **PEP use in the past 12 months**$^{**}$ | 1,530 obs | 0.971 | 1,512 obs | 0.888 |
| No | Ref | | Ref | |
| Yes | 1.04 (0.12–8.90) | | 1.16 (0.13–10.11) | |
| **PrEP use in the past 12 months**$^{**}$ | 1,532 obs | 0.97 | 1,512 obs | 0.999 |
| No | Ref | | Ref | |
| Yes | 0.96 (0.12–7.81) | | 0.99 (0.13–7.51) | |
| **Bacterial STI diagnoses**$^{\ddagger}$ | 3,819 obs | **0.005** | 3,770 obs | **0.002** |
| No | Ref | | Ref | |
| Yes | **4.46 (1.57–12.68)** | | **5.93 (1.95–18.03)** | |
| **Health and lifestyle characteristics** | | | | |
| **Recreational drug use in the past 3 months**$^{**}$ | 1,536 obs | 0.152 | 1,518 obs | 0.111 |
| No | Ref | | Ref | |
| **Yes** | 4.81 (0.56–41.26) | | 5.83 (0.66–50.97) | |
| **Chemsex in the past 3 months** | 3,819 obs | **0.012** | 3,770 obs | **0.006** |
| No | Ref | | Ref | |
| Yes | **3.89 (1.35–11.22)** | | **4.81 (1.57–14.74)** | |
| **Injection drug use in the past 3 months**$^{**}$ | 1,536 obs | **<0.001** | 1,518 obs | **0.001** |
| No | Ref | | Ref | |
| Yes | **21.67 (3.96–118.30)** | | **18.99 (3.39–106.14)** | |
| **Higher-risk alcohol consumption**$^{**}$ **(modified WHO AUDIT-C equals ≥6)** | 1,536 obs | 0.26 | 1,521 obs | 0.335 |
| No | Ref | | Ref | |
| Yes | 2.10 (0.58–7.63) | | 1.91 (0.51–7.08) | |
| **Depressive symptoms**$^{**}$ **(PHQ-9 score ≥10)** | 1,536 obs | 0.818 | 1,521 obs | 0.701 |
| No | Ref | | Ref | |
| Yes | 1.28 (0.15–10.64) | | 1.53 (0.17–13.31) | |
| **Anxiety symptoms**$^{**}$ **(GAD-7 score ≥10)** | 1,537 obs | 0.559 | 1,526 obs | 0.681 |
| No | Ref | | Ref | |
| Yes | 1.88 (0.23–15.60) | | 2.06 (0.23–18.30) | |
| **Calendar year as a continuous variable** | 3,821 obs | **<0.001** | 3,769 obs | **0.004** |
| | **0.52 (0.45–0.59)** | | **0.47 (0.28–0.78)** | |
| **Calendar year category** | 3,821 obs | 0.053 | 3,769 obs | **0.01** |
| 2013–2014 | Ref | | Ref | |
| 2015 | 0.26 (0.07–1.05) | | **0.17 (0.04–0.82)** | |
| 2016 | 0.35 (0.10–1.19) | | **0.20 (0.05–0.83)** | |

(*Continued*)

**Table 3.** (Continued)

|  | Unadjusted IRR (95% CI) | *p*-value | Adjusted€ IRR (95% CI) | *p*-value |
|---|---|---|---|---|
| 2017–2018 | **0.06 (0.01–0.54)** |  | **0.05 (0.01–0.44)** |  |

*Total complete observations: 3,821 questionnaires; sexual/HIV-related behaviour data were based on the last time man asked; number of new sexual partners, fisting or sex toys, sex for drugs or money, and smoking status were not included in the analysis because they were only asked at the baseline questionnaire.

€Adjusted for age (time-updated), country of birth and ethnicity, sexual identity, and university education.

**Data were not collected at the 4-monthly questionnaire (only baseline and annual questionnaires).

†In the past 6 months at the baseline questionnaire and in the past 3 months at the 4-monthly and annual questionnaires.

‡In the past 12 months at the baseline questionnaire and in the past 3 months at the 4-monthly and annual questionnaires.

CI, confidence interval; CLS, condomless anal sex; GAD-7, generalised anxiety disorder-7; GBMSM, gay, bisexual, and other men who have sex with men; IRR, incidence rate ratio; PEP, postexposure prophylaxis; PHQ-9, patient health questionnaire-9; PrEP, preexposure prophylaxis; STI, sexually transmitted infection; WHO-AUDIT, World Health Organization–Alcohol Use Disorders Identification Test.

"regression to the mean" effect may have operated because the men were recruited at a time of particularly high STI risk [19]. In contrast to these trends in STIs, group sex, and drug use overall, we observed in this study that the prevalence of CLS with 2 or more partners slightly increased, and the prevalence of injection drug use and noninjection chemsex-related drug use remained relatively stable, between 2013 and 2018 (baseline–the end of follow-up). The decline in HIV incidence is, therefore, unlikely to be solely explained by changes in sexual behaviour during this period.

Lower levels of infectious HIV in the community due to more timely HIV diagnosis and earlier treatment among those accessing HIV care are likely to have had a role in declining incidence, in line with previous prediction [20,21]. A recent study in Australia, the TAIPAN study, has demonstrated that the decrease in community-level HIV viraemia ($\geq$200 copies/mL) from 28.6% in 2012 to 12.8% in 2017 among HIV–positive gay and bisexual men was significantly associated with decreasing HIV incidence in New South Wales and Victoria (from 0.88 per 100 PY in 2012 to 0.22 per 100 PY in 2017) [5].

PrEP use during follow-up may also have impacted on declining HIV incidence. An important finding in our study was that the fall in HIV incidence coincided with a major increase in the proportion of men reporting past 12-month PrEP use over time [22], which could indicate an association. In our study, baseline and longitudinal reported PrEP use was not associated with reduced HIV incidence. At baseline, only 5% of men reported PrEP use in the past 12 months, and possibly, these men were early PrEP takers having high-risk sexual behaviour putting them at particularly high risk of HIV infection. It is possible that no clear association was observed due to opposing factors operating—PrEP use decreasing the risk of HIV acquisition on the one hand, and PrEP use acting as an indicator of very high-risk behaviour (similar to the other markers of CLS) on the other. Moreover, in this study, past 12-month PrEP use was only asked at baseline and annual questionnaires; therefore, we do not have a complete picture of PrEP use during follow-up, or of adherence or consistency in using PrEP. Taken together, our results are consistent with the hypothesis that the benefits of ART in reducing HIV transmission in combination with increased uptake of PrEP has had a substantial impact in reducing HIV incidence in the GBMSM population.

Recreational drug use was one of the strongest factors associated with HIV incidence in this cohort. HIV incidence was especially high among men who reported the use of injection drugs, 4.8 per 100 PY, almost 28-fold higher than the incidence among men who did not report any recreational drugs. The use of noninjection chemsex-related drugs also increased the risk of HIV incidence more than 6-fold. A systematic review investigating recreational

drug use in GBMSM has demonstrated that chemsex use is associated with increased risky behaviour such as CLS and group sex, as well as with an increase in STIs and poor mental health symptoms [23]. Polydrug use has also been reported to be associated with condomless sex and higher partner numbers in HIV–negative and HIV–diagnosed GBMSM in the UK [24,25]. There are limited data on injection drug use among GBMSM in the UK and Europe. Findings from the 2014 Gay Men's Sex Survey, an online survey of 14,464 GBMSM living in the UK, suggest that injection drug use (amphetamine, crystal methamphetamine, heroin, mephedrone, GHB/GBL, and ketamine) is significantly associated with CLS with multiple partners [26]. The survey also found that injecting was most common among those who were of age 30 to 59 years, lived in London, and were HIV seropositive. Data from Australian and Canadian GBMSM cohorts have also observed strong associations between injecting drugs and sexual risk behaviours [27,28]. Further research into the barriers to accessing HIV prevention services among GBMSM who inject drugs, despite the availability of harm reduction programmes in the UK, will be useful.

We also observed that the risk of acquiring HIV was higher among GBMSM who reported high-risk sexual behaviours (CLS with multiple or HIV–positive partners, group sex, greater number of new sexual partners, versatile CLS role, and sex for drugs or money) and bacterial STI diagnoses. Risk was particularly high for men reporting group sex and those with higher numbers of CLS partners in the past 3 months. This is consistent with findings from other cohort studies in the UK and other countries [8–9,29]. Routine inquiry and documentation of these factors could enable better direction of prevention efforts at both the individual and population level.

In the AURAH2 cohort, most demographic and socioeconomic factors were not associated with incident HIV. However, we observed a higher IR among men with nonuniversity level of education that might be explained by the higher prevalence of high-risk sexual behaviours in this subgroup of men. The prevalence of past 3-month CLS at baseline was significantly higher among men with no educational qualifications, at 86% ($p$ = 0.038), compared to men with university-level education and other qualification (**S3 Table**). A lower educational level has been reported to be associated with risk-taking behaviours and with an increased risk of HIV seroconversion in European studies [30,31]. We did not find evidence that high alcohol use, smoking, or symptoms of depression or anxiety were associated with incident HIV in the baseline associated factors or time-updated analysis, although CIs were wide for some factors. It has previously been reported that the relationship of mental health symptoms with sexual behaviour may be complex and operate in both directions [32].

The strengths of this study include the prospective design and HIV status confirmation of all 1,162 participants enrolled in AURAH2 through linkage with national HIV surveillance data. This allows for optimum use of available information to estimate HIV incidence and trends for all men in the cohort. Prior to data linkage, we have presented our interim results restricted to men under follow-up with questionnaire [33–34], adopting the single random point method to decide HIV infection dates between self-reported first HIV positive test results and last HIV negative test results [35]. We also observed significant decline over time among these men; however, trends were only able to be calculated from 2015 until 2018 (online follow-up period), and we missed a number of diagnosis that were further identified after linking our data with PHE.

There are some limitations to this study. Men in this cohort were recruited from sexual health clinics in urban areas of London and Brighton and are predominantly highly educated, employed, in a stable economic situation, and of white ethnicity. These men may not be representative of the broader GBMSM population in England and the UK. It is possible that the incidence estimates and risk factors identified are not generalizable to GBMSM

who do not attend sexual health clinics. The small number of HIV infections in each calendar year among men in this study has resulted in relatively wide CIs of IRs; therefore, IRs and associations with factors must be interpreted carefully. In addition, assessment of trends over time in sexual behaviour may be subject to "regression to the mean" as the clinic visit at which recruitment occurred may have been specifically prompted by a recent period of higher risk. For risk factors analysis, we focused on baseline factors in order to include all data from the whole cohort, which may have underestimated the associations between sexual/HIV-related behaviours and HIV incidence, including the impact of PrEP. However, we observed similar results when analysis was restricted to 622 men using time-updated variables. In terms of the time-updated analysis, the online retention of participants who initially registered in the study was not optimal; however, 64% (400 of 622) of participants who completed at least an online questionnaire were engaged in the study throughout. Our results may be sensitive to specific recall bias and social desirability bias in men's responses in the baseline questionnaire. Data linkage to surveillance systems using pseudo-anonymised identifiers has potential for mismatches or missing seroconversions; however, this has been minimised by PHE data triangulation; all self-reported seroconversions were validated by PHE data. Lastly, this study would not include seroconversions that were not diagnosed or those that were diagnosed outside the UK.

In summary, this study provides evidence of a substantial decline in HIV incidence among a cohort of GBMSM attending sexual health clinics in England. Our data suggest that GBMSM reporting the use of recreational drugs, in particular injection drug use and chemsex drug use, high-risk sexual behaviours such as CLS with multiple partners, CLS with HIV–positive partners, group sex, and those with a bacterial STI, are at increased risk of HIV acquisition. HIV infections are also significantly higher among those with lower levels of education at baseline. Temporal trends in sexual risk behaviours and drug use in the cohort over the study period were mixed, but the marked decrease in incidence coincided with a substantial increase in PrEP use. Given similar findings from recent data among GBMSM in the UK and other countries, it is likely that the observed decline is largely related to the increase in testing and earlier ART initiation from 2013 onward and the scale-up of PrEP. Although efforts to end HIV epidemic are having a substantial effect, further improvements specially to increase HIV test coverage across all populations at risk remain very important. Sustainable and comprehensive HIV prevention and control efforts must continue in the UK to reach zero new infections by 2030.

## Supporting information

**S1 Checklist. Strengthening the Reporting of Observational Studies in Epidemiology (STROBE) Statement The AURAH2 Study.**
(DOC)

**S1 Analyses Plan. Longitudinal analysis of new HIV infections and their predictors among MSM in England: The AURAH2 Study–Data Analysis Plan.**
(PDF)

**S1 Table. Baseline sociodemographic, health and lifestyle characteristics, sexual behaviour, and PrEP and PEP use among participants who completed the baseline, 4-monthly, and annual questionnaire in the AURAH2 study.**
(DOCX)

**S2 Table. Adjusted associations of baseline characteristics with incident HIV among 1,162 GBMSM participating in the AURAH2 study.**
(DOCX)

**S3 Table. Associations between ethnicity, education, and employment characteristics with sexual behaviour measures at baseline among 1,162 GBMSM in the AURAH2 study.** (DOCX)

## Acknowledgments

We thank all the study participants for their time and effort. The AURAH2 Study Group acknowledges the support of the NIHR through the Comprehensive Clinical Research Network. The AURAH2 study was also sponsored by the Joint Research Office, UCL.

**The members of The AURAH2 Study Group are the following:** Alison J. Rodger, Fiona C. Lampe, Andrew N. Phillips, Valentina Cambiano, Janey Sewell, Andrew Speakman, Ada R. Miltz, Nadia Hanum, Richard Gilson, Nneka Nwokolo, Amanda Clarke, David Asboe, Simon Collins, Ana Milinkovic, Fabienne Styles, Rosanna Laverick, Marzena Orzol, Emmi Suonpera, Ali Ogilvy, Celia Richardson, Elaney Youssef, Sarah Kirk, Marion Campbell, and Lisa Barbour.

**Disclaimer**

The views expressed in this study are those of the author(s) and not necessarily those of the NIHR or the Department of Health and Social Care.

## Author Contributions

**Conceptualization:** Nadia Hanum, Valentina Cambiano, Janey Sewell, Alison J. Rodger, Nneka Nwokolo, David Asboe, Richard Gilson, Amanda Clarke, Ada R. Miltz, Simon Collins, Valerie Delpech, Sara Croxford, Andrew N. Phillips, Fiona C. Lampe.

**Data curation:** Nadia Hanum, Valentina Cambiano, Janey Sewell, Ada R. Miltz.

**Formal analysis:** Nadia Hanum.

**Methodology:** Alison J. Rodger, Andrew N. Phillips, Fiona C. Lampe.

**Writing – original draft:** Nadia Hanum.

**Writing – review & editing:** Nadia Hanum, Valentina Cambiano, Janey Sewell, Alison J. Rodger, Nneka Nwokolo, David Asboe, Richard Gilson, Amanda Clarke, Ada R. Miltz, Simon Collins, Valerie Delpech, Sara Croxford, Andrew N. Phillips, Fiona C. Lampe.

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
