## [Editor Report · Decision Letter 0]

3 Mar 2021

Dear Dr Hanum, 

Thank you for submitting your manuscript entitled "Trends in HIV incidence between 2013 – 2019 and baseline predictors among a prospective cohort of Gay, Bisexual, and Other Men Who Have Sex with Men attending sexual health clinics in England: the AURAH2 Study" for consideration by PLOS Medicine.

Your manuscript has now been evaluated by the PLOS Medicine editorial staff and I am writing to let you know that we would like to send your submission out for external peer review.

Please re-submit your manuscript within two working days, i.e. by March 5, 2021.

Kind regards,

Beryne Odeny

Associate Editor

PLOS Medicine

---

## [Decision Letter · Decision Letter 1]

9 Apr 2021

Dear Dr. Hanum,

Thank you very much for submitting your manuscript "Trends in HIV incidence between 2013 – 2019 and baseline predictors among a prospective cohort of Gay, Bisexual, and Other Men Who Have Sex with Men attending sexual health clinics in England: the AURAH2 Study" (PMEDICINE-D-21-01009R1) for consideration at PLOS Medicine. 

[LINK]

In light of these reviews, I am afraid that we will not be able to accept the manuscript for publication in the journal in its current form, but we would like to consider a revised version that addresses the reviewers' and editors' comments. Obviously we cannot make any decision about publication until we have seen the revised manuscript and your response, and we plan to seek re-review by one or more of the reviewers. 

We expect to receive your revised manuscript by Apr 30 2021 11:59PM. Please email us (plosmedicine@plos.org) if you have any questions or concerns.

We look forward to receiving your revised manuscript. 

Sincerely,

Beryne Odeny, 

PLOS Medicine

plosmedicine.org

- Please revise your title according to PLOS Medicine's style. Your title must be nondeclarative and not a question. It should begin with main concept if possible. Please place the study design ("A prospective cohort study,") in the subtitle (i.e., after a colon). 

- The Data Availability Statement (DAS) requires revision. For each data source used in your study: If the data are not freely available, please describe briefly the ethical, legal, or contractual restriction that prevents you from sharing it. Please also include an appropriate contact (web or email address) for inquiries (again, this cannot be a study author/ co-author).

- Abstract summary - At this stage, we ask that you reformat your non-technical Author Summary. The Author Summary should immediately follow the Abstract in your revised manuscript. This text is subject to editorial change and should be distinct from the scientific abstract. The summary should be accessible to a wide audience that includes both scientists and non-scientists. Please see our author guidelines for more information: https://journals.plos.org/plosmedicine/s/revising-your-manuscript#loc-author-summary.

- Abstract:

1. Please structure your abstract using the PLOS Medicine headings (Background, Methods and Findings, Conclusions).

2. Please combine the Methods and Findings sections into one section, “Methods and findings”. Please ensure that all numbers presented in the abstract are present and identical to numbers presented in the main manuscript text.

3. Please include the actual amounts or percentages of relevant outcomes, not just hazard ratios or relative risks.

4. Please include the important dependent variables that are adjusted for in the analyses.

5. Please quantify the main results (with p values in addition to 95% CI).

6. Please include a summary of adverse events if these were assessed in the study.

7. In the last sentence of the Abstract Methods and Findings section, please describe the main limitation(s) of the study's methodology.

8. Abstract Conclusions: Please address the study implications without overreaching what can be concluded from the data; the phrase "In this study, we observed ..." may be useful. Please interpret the study based on the results presented in the abstract, emphasizing what is new.

- Please conclude the Introduction with a clear description of the study question or hypothesis.

- For this observational study, in the manuscript text, please indicate: (1) the analytical methods by which you planned to test your hypothesis, (2) the analyses you actually performed, and (3) when reported analyses differ from those that were planned, transparent explanations for differences that affect the reliability of the study's results. If a reported analysis was performed based on an interesting but unanticipated pattern in the data, please be clear that the analysis was data-driven.

- Did your study have a prospective protocol or analysis plan? Please state this (either way) early in the Methods section. 

- Please ensure that the study is reported according to the STROBE guideline, and include the completed STROBE checklist as Supporting Information. Please add the following statement, or similar, to the Methods: "This study is reported as per the Strengthening the Reporting of Observational Studies in Epidemiology (STROBE) guideline (S1 Checklist)." The STROBE guideline can be found here: http://www.equator-network.org/reporting-guidelines/strobe/

- Your study is observational and therefore causality cannot be inferred. Please remove language that implies causality, such as “predictive” or “due to”. Refer to associations instead.

- How was race/ethnicity defined and by whom? Why was race/ethnicity considered important in this study and what it is believed to represent [eg, are SES or genetic differences being attributed to race/ethnicity?]

- In your statistical analyses, please use hierarchical/ multilevel models given that nationwide data is likely clustered at various county/ regional levels. The potential clustering of data (e.g., among patients from the same locality or hospital) would result in spurious effect estimates and standard errors

- In statistical methods, please refer to any post-hoc corrections to correct for multiple comparisons during your statistical analyses. If these were not performed please justify the reasons. Please refer to our statistical reporting guidelines for assistance (https://journals.plos.org/plosone/s/submission-guidelines.#loc-statistical-reporting)

- Please describe how you selected your adjustment variables. 

- Please specify the statistical test used to derive a p value.

- Please define the abbreviations in tables and/or Figures e.g. PY, GBMSM, CLY, PHQ-9, GAD7

- Please include p-values in table 2 

- Please change p <0.0001 to P < 0.001.

- Please provide titles and legends for all figures (including those in Supporting Information files).

- Please update the access or retrieval date for all website citations to ensure that websites are still available

- Please use the "Vancouver" style for reference formatting, and see our website for other reference guidelines https://journals.plos.org/plosmedicine/s/submission-guidelines#loc-references

- Please replace “injection drug users” with “people who inject drugs.”

Comments from the reviewers:

Reviewer #1: This study reports trends in HIV incidence between 2013 and 2019 and the association of baseline and time-updated factors (demographic, socioeconomic, behavioural, lifestyle and health related) with HIV incidence, among a cohort of GBMSM in the AURAH2 prospective study.

Comments:

There are some typos in the text that need remedying.

"Supplementary material 1: Baseline socio-demographic, health and lifestyle characteristics, sexual behaviour, and PrEP and PEP use among participants who completed the baseline, four-monthly, and annual questionnaire in the AURAH2 study, 2013 - 2018"

Did the authors consider completing a comparison between groups, which may be insightful to gauge potential response bias? 

It is noted that the groups in Supplementary material 1 are not independent groups, and so the authors may choose to separately amend the structure of the groups in order to draw comparisons (baseline only compared with baseline+four-monthly follow-up, for example). 

Overall, the authors have applied technically appropriate statistical methods, and presented the study outcomes in a clear and concise way. 

The study limitations have been suitably acknowledged in the discussion section.

Reviewer #2: This is a timely analysis of changes in HIV incidence, sexual risk behavior, and PrEP use in a cohort of initially seronegative GBMSM from 3 urban STD clinics in the UK. The results are encouraging and demonstrate declining incidence as well as a decrease in some high-risk sexual behaviors, with concomitant increase in PrEP use. While there are limitations of the work, it represents an important contribution to our knowledge of changing HIV risk among GBMSM globally. I have a few queries and suggestions for the authors to consider, in order to improve clarity of the manuscript.

Queries and suggestions:

1. Methods, study design and participants. What is the justification for combining years in two of the categories (2013/2014 and 2018/2019). One can guess, but this should be explicitly stated.

2. Methods, baseline measures. Were participants asked about sexual role taking, such as top/insertive, bottom/receptive, or versatile? If not, this should be mentioned as a limitation of the questionnaires used.

3. Methods, p. 6 lines 155-157. The cutpoints for PHQ-9 and GAD-7 need a justification. This should be easy, as both are "moderate or severe" categories, but an explanation is needed for readers not familiar with these scales and their scoring. Similarly, the selected cutpoint for AUDIT-C needs justification and a reference.

4. Methods, p. 6, lines 158-160. The handling of missing data needs more justification. Why considering missing responses as "no" responses for behavior, mental health, and alcohol consumption and other variables as missing = excluded? Were data likely to be missing at random? Was imputation considered as an option to reduce bias due to missingness?

5. Results, lines 233-235. It is said that 622 of 1162 completed at least one online follow-up questionnaire, of whom only 483 completed at least one online follow-up questionnaire. How is this possible? Please edit for clarity. It looks like 622 were included in the analysis of time-updated variables.

6. Discussion, PrEP use. How might an approach combining PrEP and CLS to develop an "unprotected sex" variable change results. Did PrEP use match risk in this study or was it underutilized by some groups?

7. Discussion, injection drug use. The degree to which risk is increased with injection drug use in this cohort is noteworthy. Do GBMSM who inject drug have access to needle exchange and opioid substitution therapy in the UK? Is there any information on the extent to which men in this cohort accessed such programs?

8. Discussion, limitations, lines 416-418. Please clarify the writing with respect to the 400 participants who are said to have competed "at least an online questionnaire" and were engaged in the study throughout. How does this relate to the earlier statements that 622 of 1162 completed at least one online follow-up questionnaire, of whom only 483 completed at least one online follow-up questionnaire.

9. Discussion, limitations. Could the authors be more specific about how the sociodemographics of the men participating in this cohort compare to those of new HIV cases among GBMSM in the UK? Which specific groups of men are underrepresented and warrant more targeted approaches? It would be nice to see some discussion on how this work relates to current known disparities in access to HIV services (for both positives and negatives) in the UK. Where might treatment as prevention and PrEP be falling short of their promise?

Minor suggested corrections or edits:

1. General. Consider whether condomless anal sex is best abbreviated "CLAS" or "CAS" rather than "CLS."

2. Methods, line 159. "missing responses were considered to indicate that the absence…" Remove "that."

3. Methods, line 189. "were include" should be "were included."

4. Results, line 227. "taken at least on chemsex-related drugs" needs editing.

5. Table 1, country of birth and ethnicity. Suggest changing "Born on the UK, other ethnicity" to "Born in the UK, other ethnicity.

6. Table 1 footnotes. Consider changing from "No group includes…" to "The no employment group includes…"

7. Supplementary Table 2. For consistency, CLS in the past 3 months should be bolded.

8. Figure 2 caption. It is said in the text that annual questionnaire data were excluded for bacterial STI due to a difference in recall periods. Please correct the Figure 2 caption to reflect this.

Reviewer #3: Linda Anne Selvey

Thanks for this manuscript, which has some useful information that confirms the findings of others. I have a number of comments/suggestions below:

1. Methods

More detail is required in the methods in relation to the analysis, particularly in terms of missing values. With missing values in the multivariable analysis, was the whole line (ie data about a single individual) excluded from the analysis, or were the missing values handled in a different way. This needs to be explicit in the methods. The total number of observations in the final model should also be included in the results section, not just in table 3. 

Also in the methods there is a description of the categorisation of the CLS and new sexual partners variables. How were these classifications made and why were these categories chosen. This detail should also be included. Zero to 10 new sexual partners in the last 12 months seems like a wide category. Why not have a variable with just Zero or one new partners?

2. Editing

The paper needs a bit of an edit. For example line 189 (include vs included); line 227 (on vs one); line 236 (no full stop after 'study')

3. Discussion

There were very small numbers of HIV infections in each calendar year, therefore with wide confidence intervals. This should be included as a limitation in the discussion.

[LINK]

---

## [Decision Letter · Decision Letter 2]

14 May 2021

Dear Dr. Hanum,

Thank you very much for re-submitting your manuscript "Trends in HIV incidence between 2013 – 2019 and baseline predictors among a prospective cohort of Gay, Bisexual, and Other Men Who Have Sex with Men attending sexual health clinics in England: A prospective cohort study" (PMEDICINE-D-21-01009R2) for review by PLOS Medicine.

I have discussed the paper with my colleagues and the academic editor and it was also seen again by one reviewer. I am pleased to say that provided the remaining editorial and production issues are dealt with we are planning to accept the paper for publication in the journal.

[LINK]

We look forward to receiving the revised manuscript by May 21 2021 11:59PM.   

Sincerely,

Beryne Odeny, 

Associate Editor 

PLOS Medicine

plosmedicine.org

Requests from Editors:

1. In the author summary, under the sub-heading, “what do these findings mean?”, please remove language that implies attribution or cause. Use phrases such as “may potentially be attributed to…” In the second bullet point replace the word “should” with “may potentially” or “could” or similar.

2. Please replace “injection drug users” with “people who inject drugs.” For example, line 272.

3. Please remove the “&” in reference # 11.

Comments from Reviewers:

Reviewer #2: The revision has addressed prior comments, and I believe the article is acceptable in its present form. 

However, I would also suggest that the authors consider reducing the numeric results in the abstract, dropping either the numerators and denominators, the p values, or both, to make the abstract findings more readable and potentially include some of the other interesting findings, such as the increase in PrEP use, decrease in PEP and some risk behaviors, but lack of association of PrEP with the outcome.

I also note two very minor edits that are needed, below.

Minor edits:

1. Abstract: In the very long sentence on baseline associations, an extra "and" appears before "previous 12-month report of a bacterial STI diagnosis" and there is no HR with 95% CI provided for the finding on new sexual partners.

2. on p.14 under table 2, the sentence starting "90% men" needs an edit. Perhaps "The vast majority (27/30 or 90%) of men newly diagnosed with HIV were of White ethnicity."?

[LINK]

---

## [Editor Report · Decision Letter 3]

25 May 2021

Dear Dr. Hanum,

Thank you very much for re-submitting your manuscript "Trends in HIV incidence between 2013 – 2019 and baseline predictors among a prospective cohort of Gay, Bisexual, and Other Men Who Have Sex with Men attending sexual health clinics in England: A prospective cohort study" (PMEDICINE-D-21-01009R3) for review by PLOS Medicine.

I have discussed the paper with my colleagues and the academic editor and it was also seen again by three reviewers. I am pleased to say that provided the remaining editorial and production issues are dealt with we are planning to accept the paper for publication in the journal.

[LINK]

We look forward to receiving the revised manuscript by Jun 01 2021 11:59PM.   

Sincerely,

Beryne Odeny, 

Association Editor 

PLOS Medicine

plosmedicine.org

Final requests from Editors:

Thank you for responding to previous comments. Before we proceed, please make the following changes:

1. Please remove the first mention of "prospective cohort" from the title. Kindly consider revising the title to, "Trends in HIV incidence between 2013 – 2019 and baseline predictors among Gay, Bisexual, and Other Men Who Have Sex with Men attending sexual health clinics in England: A prospective cohort study" 

2. For consistency, please replace the term "predictors" with "factors" in the title and throughout the manuscript and tables.

Comments from Reviewers:

[LINK]

---

## [Editor Report · Decision Letter 4]

1 Jun 2021

Dear Dr Hanum, 

On behalf of my colleagues and the Academic Editor, Dr. Susan Marie Graham, I am pleased to inform you that we have agreed to publish your manuscript "Trends in HIV incidence between 2013 – 2019 and association of baseline factors with subsequent incident HIV among Gay, Bisexual, and Other Men Who Have Sex with Men attending sexual health clinics in England: A prospective cohort study" (PMEDICINE-D-21-01009R4) in PLOS Medicine.

PRESS

Sincerely, 

Beryne Odeny 

Associate Editor 

PLOS Medicine